# How the Severity and Mechanism of Recurrent Laryngeal Nerve Dysfunction during Monitored Thyroidectomy Impact on Postoperative Voice

**DOI:** 10.3390/cancers13215379

**Published:** 2021-10-27

**Authors:** Tzu-Yen Huang, Wing-Hei Viola Yu, Feng-Yu Chiang, Che-Wei Wu, Shih-Chen Fu, An-Shun Tai, Yi-Chu Lin, Hsin-Yi Tseng, Ka-Wo Lee, Sheng-Hsuan Lin

**Affiliations:** 1International Thyroid Surgery Center, Department of Otolaryngology—Head and Neck Surgery, Kaohsiung Medical University Hospital, Faculty of Medicine, College of Medicine, Kaohsiung Medical University, Kaohsiung 807, Taiwan; tyhuang.ent@gmail.com (T.-Y.H.); kmuhentst@gmail.com (W.-H.V.Y.); cwwu@kmu.edu.tw (C.-W.W.); reddust0113@yahoo.com.tw (Y.-C.L.); sycatlovestar@gmail.com (H.-Y.T.); kawolee@kmu.edu.tw (K.-W.L.); 2Department of Biological Science and Technology, National Yang Ming Chiao Tung University, Hsinchu 300, Taiwan; 3Department of Otolaryngology—Head and Neck Surgery, E-Da Hospital, Kaohsiung 824, Taiwan; fychiang@kmu.edu.tw; 4School of Medicine, College of Medicine, I-Shou University, Kaohsiung 824, Taiwan; 5Department of Otolaryngology—Head and Neck Surgery, Kaohsiung Municipal Siaogang Hospital, Kaohsiung Medical University Hospital, Faculty of Medicine, College of Medicine, Kaohsiung Medical University, Kaohsiung 812, Taiwan; 6Institute of Statistics, National Yang Ming Chiao Tung University, Hsinchu 300, Taiwan; fushihchen@gmail.com (S.-C.F.); daansh13@gmail.com (A.-S.T.); 7Department of Otolaryngology—Head and Neck Surgery, Kaohsiung Municipal Tatung Hospital, Kaohsiung Medical University Hospital, Faculty of Medicine, College of Medicine, Kaohsiung Medical University, Kaohsiung 801, Taiwan; 8Institute of Data Science and Engineering, National Yang Ming Chiao Tung University, Hsinchu 300, Taiwan

**Keywords:** recurrent laryngeal nerve (RLN), intraoperative neuromonitoring (IONM), severity and mechanism of RLN dysfunction, postoperative voice, index of voice and swallowing handicap of thyroidectomy (IVST)

## Abstract

**Simple Summary:**

Recurrent laryngeal nerve (RLN) dysfunction remains a major source of morbidity after thyroid surgery. Intraoperative neuromonitoring can qualify and quantify RLN function according to the laryngeal electromyography (EMG) response evoked by electrical stimulation of the RLN. To the best of our knowledge, this is the first report to discuss the severity and mechanism of RLN dysfunction and postoperative voice in patients who have received monitored thyroidectomy. For optimal voice and swallowing outcomes after thyroid surgery, thermal injury must be avoided, especially when using energy-based devices, and mechanical injury must be identified early to avoid a more severe dysfunction. Adherence to standard intraoperative neuromonitoring (IONM) procedures for thyroid surgery is suggested, including standard procedures for acquiring and interpreting intraoperative RLN signals, for identifying and classifying RLN injury mechanisms, for performing laryngeal examinations and comprehensive voice assessments (subjective and objective voice analysis) before and after surgery, and for performing standard follow-up procedures.

**Abstract:**

Intraoperative neuromonitoring can qualify and quantify RLN function during thyroid surgery. This study investigated how the severity and mechanism of RLN dysfunction during monitored thyroid surgery affected postoperative voice. This retrospective study analyzed 1021 patients that received standardized monitored thyroidectomy. Patients had post-dissection RLN(R2) signal <50%, 50–90% and >90% decrease from pre-dissection RLN(R1) signal were classified into Group A-no/mild, B-moderate, and C-severe RLN dysfunction, respectively. Demographic characteristics, RLN injury mechanisms(mechanical/thermal) and voice analysis parameters were recorded. More patients in the group with higher severity of RLN dysfunction had malignant pathology results (A/B/C = 35%/48%/55%, *p* = 0.017), received neck dissection (A/B/C = 17%/31%/55%, *p* < 0.001), had thermal injury (*p* = 0.006), and had asymmetric vocal fold motion in long-term postoperative periods (A/B/C = 0%/8%/62%, *p* < 0.001). In postoperative periods, Group C patients had significantly worse voice outcomes in several voice parameters in comparison to Group A/B. Thermal injury was associated with larger voice impairments compared to mechanical injury. This report is the first to discuss the severity and mechanism of RLN dysfunction and postoperative voice in patients who received monitored thyroidectomy. To optimize voice and swallowing outcomes after thyroidectomy, avoiding thermal injury is mandatory, and mechanical injury must be identified early to avoid a more severe dysfunction.

## 1. Introduction

Recurrent laryngeal nerve (RLN) dysfunction remains a major source of morbidity after thyroid surgery. The RLN dysfunction during thyroid surgery can cause vocal fold paralysis (VFP), which interferes with voice and can potentially interfere with breathing and cause aspiration [1,2]. The RLN innervates the intrinsic laryngeal muscles, which control vocal fold motion (VFM). A motor unit is a single nerve fiber and all muscle fibers are innervated by it; numerous motor units must be activated simultaneously for effective muscle contraction. If excessive RLN nerve fibers are injured during thyroid surgery, their dysfunction and lack of participation in polarization can cause postoperative VFP [3,4]. A major development in thyroid surgery in the past decade is the widespread use of intraoperative neuromonitoring (IONM) to identify the RLN adjunct to the standard practice of visual identification. Use of IONM enables surgeons to qualify and quantify neural function in real time by observing the laryngeal EMG response evoked by electrical stimulation of the RLN or the vagus nerve (VN) [5,6,7,8]. Partial or total loss of nerve conduction observed during nerve stimulation indicates the occurrence of RLN stress or injury. The surgeon can also use IONM to determine what surgical maneuver caused the impending or actual RLN injury. By elucidating RLN injury mechanisms and surgical pitfalls, IONM can assist surgeons in improving their surgical techniques, in predicting recovery outcomes, and in planning intra- and post-operative management [9,10,11,12].

The severity of dysphonia depends on neural remaining function, neural recovery status, and muscular compensation mechanisms. Many researchers have discussed the widely varying methods of evaluating RLN dysfunction and voice outcomes after thyroid surgery [13,14,15,16]. After thyroidectomy, voice performance may be impaired even in patients without postoperative VFP [17,18] and in patients with RLN visual integrity [19]. To minimize bias in reporting thyroidectomy outcomes, Jeannon et al. [20] suggested establishing standardized postoperative assessments of RLN dysfunction and postoperative voice. The standardized procedures performed in this study mainly referred to International Neural Monitoring Study Group (INMSG) guidelines [6,21,22]: (1) quantification of the severity of RLN dysfunction according to the change in RLN electromyography (EMG) signals recorded intraoperatively; (2) qualification of RLN dysfunction mechanisms according to RLN injury mechanisms revealed by EMG signals change; and (3) performance of standard pre- and post-operative laryngeal examinations and voice assessments (subjective and objective voice analyses) in all patients during routine follow up.

To the best of our knowledge, no studies have investigated the post-thyroidectomy voice by using IONM to qualify RLN dysfunction mechanisms and to quantify RLN dysfunction severity. Therefore, this study evaluated how the severity and mechanism of RLN dysfunction during monitored thyroid surgery affects postoperative subjective/objective voice performance.

## 2. Materials and Methods

This study retrospectively enrolled 1119 patients who had received IONM-assisted thyroid surgery performed by the “IONM team” at Kaohsiung Medical University Hospital from January 2013, to December 2019. Patients were excluded if they were younger than 18 years (*n* = 26) or had preoperative asymmetric VFM (*n* = 18). Additionally, patients were excluded if they had any history of neck surgery, neck irradiation, or head and neck malignancy other than thyroid cancer (*n* = 54). Figure 1 is a flowchart of the inclusion and exclusion of patients in this study.

Ethical approval of this study was obtained from the Kaohsiung Medical University Hospital Institutional Review Board (KMUHIRB-E(I)-20200131). In all patients, routine evaluation of VN function and RLN function was performed under IONM using the standard four-step (V1-R1-R2-V2) procedure. All EMG amplitudes (V1-R1-R2-V2) were obtained and recorded when VN stimulation before thyroid dissection (V1) and after thyroidectomy (V2); and RLN stimulation at initial identification (R1) and after dissection (R2) [7,23]. The severity of RLN dysfunction was defined as the EMG amplitude decrease from pre-dissection R1 signal to post-dissection R2 signal. In the patients who had received bilateral thyroidectomy, none had an RLN signal decrease >50% on both sides. The patients were classified into three groups according to the RLN side with the most severe dysfunction: Group A included patients had no/mild RLN dysfunction with <50% RLN signal decrease; Group B included patients had moderate RLN dysfunction with 50–90% RLN signal decrease; Group C included patients had severe RLN dysfunction with >90% RLN signal decrease.

All exposed RLNs and all visually identifiable injury locations (mechanical or thermal) were documented by photograph, and using IONM before and after all the mechanical neural dissections and EBD activations to instantly determine the injury mechanism of RLN. In each RLN, stimulation was performed from the most proximal to the most distal ends of the RLN to localize the injured point. Mechanical injury was classified as type 1 (segmental injury) or type 2 (global injury) according to the criteria proposed by Chiang et al. [7] In all patients, laryngofiberscopy was documented by video before surgery and two weeks after surgery. In patients with asymmetric VFM after surgery, additional examinations were performed monthly until recovery. Patients who had asymmetric VFM 3 months after surgery would be referred to voice interventions, and the follow-up time will depend on the intervention patient received. No patients in this study had received RLN reinnervation or nerve grafting within three months of surgery.

Patient information, including gender, age, surgery type (unilateral versus bilateral), pathology results (benign versus malignant), central neck dissection (CND) for level VI and/or level VII cervical lymph nodes, and lateral neck dissection (LND) for indicated level II to level V cervical lymph nodes, was recorded and compared between groups.

### 2.1. Objective and Subjective Voice Analyses

Subjective and objective voice analyses were performed in all patients in four periods: period-I (preoperative period, within 2 months before surgery), period-II (immediate postoperative period, median duration of three days, range of one to seven days), period-III (short-term postoperative period, median duration of 12 days, range of seven–30 days); period-IV (long-term postoperative period, median duration of 40 days, range of 30–90 days).

All objective voice analysis was performed by a single experienced speech-language pathologist (WHV. Y.). The Multidimensional Voice Program (model 5105, version 3.1.7; KayPENTAX, Montvale, NJ, USA) results included mean fundamental frequency (Mean F0), jitter, shimmer and noise-to-harmonic ratio (NHR). The Voice Range Profile (model 4326, version 3.3.0; KayPENTAX, Montvale, NJ, USA) results included maximum pitch frequency (Fmax), minimum pitch frequency (Fmin), and pitch range (PR). The PR was defined as the number of semitones between Fmax and Fmin.

The subjective voice analysis was evaluated by the Index of Voice and Swallowing Handicap of Thyroidectomy (IVST) (Appendix A). The IVST was designed according to the main symptoms observed before and after thyroid surgery. Each of the 10 questionnaire items in this subjective assessment is assigned a score of 0 (never), 1 (sometimes), or 2 (always). The voice domain (IVST-V) includes items 1–7 and has a score range of 0–14. The swallowing domain (IVST-S) includes items 8–10 and has a score range of 0–6. Thus, the total IVST score (IVST-T) has a score range of 0–20.

Where A and B are preoperative and postoperative values, respectively, the equation for calculating postoperative change in objective voice analysis data was Δ = (B − A)/A; the equation for calculating postoperative change in subjective voice analysis data was Δ = B − A.

### 2.2. Statistical Analysis

To analyze the variables, an independent *t* test, a Pearson chi-square test, and an ANOVA test were performed using R software (version-3.4, R Foundation for Statistical Computing, Vienna, Austria). A two-tailed *p* value less than 0.05 was considered statistically significant.

## 3. Results

### 3.1. Demographic Characteristics of Patients

This study analyzed 1021 patients who had received primary thyroid surgery. Table 1 indicates that, when the patients were grouped according to the RLN side with the highest severity of dysfunction, Group A had 940 (92.1%) patients, Group B had 52 (5.1%) patients, and Group C had 29 (2.8%) patients. Age and gender did not significantly differ between groups. The proportion of patients who had received bilateral surgery was 73.0%, 84.6%, and 96.6% in Group A, B and C, respectively. Significantly more bilateral surgery in the group with higher severity of RLN dysfunction (*p* = 0.004). The proportion of patients that had a malignant pathology report was 35.1%, 48.1%, and 55.2% in Group A, B and C, respectively. There were significantly more malignant pathology reports in the group with higher severities of RLN dysfunction (*p* = 0.017).

In Group A, B, and C, the proportions of patients who had received CND without LND were 14.5%, 19.2%, and 44.8%, respectively, whereas the proportions of patients in Group A, B, and C who had received both CND and LND were 2.3%, 11.5%, and 10.4%, respectively. Significantly more neck dissection were performed in the group with higher severities of RLN dysfunction (*p* < 0.001).

In Group A, all exposed 1626 recurrent laryngeal nerves at risk (NARs) were visually intact, and there were no mechanical or thermal injured areas with EMG loss could be mapped. In the 96 NARs in Group B, 51 NARs had mechanical injury, 43 NARs had type 1 mechanical injury, eight NARs had type 2 mechanical injury, and one NAR had a thermal injury. In the 29 NARs in Group C, 22 NARs had mechanical injury, 17 NARs had type 1 mechanical injury, five NARs had type 2 mechanical injury, and seven NARs had thermal injuries. The proportion of NARs with thermal injury was significantly larger in Group C compared to Group B (*p* = 0.006); there was no significant difference in the type of mechanical injury between groups (*p* = 0.521).

In Group A, all patients had symmetric VFM after surgery. In Group B, 25.0% had asymmetric VFM during period-III, and 7.7% had asymmetric VFM during period-IV. In Group C, 93.1% had asymmetric VFM during period-III, and 62.1% had asymmetric VFM during period-IV. That is, asymmetric VFM was significantly associated with high severity of RLN dysfunction (*p* < 0.001).

### 3.2. Voice Parameters in Patients with Different Severity of RLN Dysfunction

Figure 2 compares voice parameters in Group A, B, and C in each follow-up period, and Appendix A gives the *p* values and additional details. In period-I, none of the objective and subjective voice parameters significantly differed between groups. In comparing Group A and Group B, only the pitch range (PR) of patients in Group B was significantly lower than those in Group A in period-II (*p* = 0.030), no other voice parameters showed significant differences between the two groups. In all periods, Fmin, Fmax, Mean F0, and NHR did not significantly differ between groups.

In comparisons of Group A and Group C, significant differences were as follows: PR in period-II (*p* = 0.001) and period-III (*p* = 0.003); jitter in period-II (*p* < 0.001), period-III (*p* < 0.001), and period-IV (*p* = 0.048); shimmer in period-III (*p* = 0.040); IVST-T in period-II (*p* < 0.001), period-III (*p* < 0.001), and period-IV (*p* = 0.025); IVST-V in period-II (*p* < 0.001), period-III (*p* < 0.001), and period-IV (*p* = 0.026); and IVST-S in period-II (*p* < 0.001) and period-III (*p* = 0.045).

In comparisons of Group B and Group C, significant differences were as follows: Jitter in period-II (*p* < 0.001) and period-III (*p* = 0.010); IVST-T in period-II (*p* < 0.001), period-III (*p* < 0.001), and period-IV (*p* = 0.046); IVST-V in period-II (*p* < 0.001), period-III (*p* < 0.001), and period-IV (*p* = 0.043); and IVST-S in period-II (*p* < 0.001) and period-III (*p* = 0.031).

### 3.3. Voice Parameter Changes (Δ) with Different Mechanism of RLN Dysfunction

Figure 3 compares voice parameter changes (Δ) between the 73 patients with mechanical injury and the 8eightpatients with thermal injuries in each follow-up period. Compared to the mechanical injury group, the thermal injury group had larger voice impairments, and significant differences were as follows: ΔFmax in period-III (*p* = 0.046) and period-IV (*p* = 0.003); ΔJitter in period-II (*p* = 0.021); ΔIVST-T in period-II (*p* = 0.042) and period-III (*p* = 0.031); ΔIVST-V in period-III (*p* = 0.021); and ΔIVST-S in period-III (*p* = 0.039).

Appendix A shows additional details of the comparisons of voice parameter changes (Δ) by mechanism of RLN injury and by severity of RLN dysfunction. Compared to patients with mechanical injury in Group B, patients with mechanical injury in Group C had larger voice impairments, and significant differences were as follows: ΔPR in period-II (*p* = 0.005) and period-III (*p* = 0.006); ΔIVST-T in period-II (*p* = 0.004); ΔIVST-V in period-II (*p* = 0.006), period-III (*p* = 0.044), and period-IV (*p* = 0.048); and ΔIVST-S in period-II (*p* = 0.001). Thermal injury in Group C showed no significant voice outcome difference compared to mechanical injury in Group C, except in ΔFmax during period-III (*p* = 0.001) and period-IV (*p* = 0.011).

## 4. Discussion

This study investigated voice outcomes in 1021 patients who had received monitored thyroidectomy. Patients with high severity of RLN dysfunction had significantly higher incidences of malignant pathology results (*p* = 0.017), neck dissection (*p* < 0.001), thermal injury (*p* = 0.006), and asymmetric VFM (*p* < 0.001) (Table 1). In period-I, no voice parameters significantly differed between groups; In postoperative periods (II/III/IV), Group C patients had significantly worse voice outcomes for several voice parameters in comparison to Group A and Group B (Figure 2). In comparisons of RLN dysfunction mechanisms, the thermal injury group had larger voice impairments than the mechanical injury group (Figure 3). Therefore, for optimal voice and swallowing outcomes after thyroidectomy, avoiding thermal injury is mandatory, and mechanical injury must be identified early to avoid a more severe dysfunction.

The postoperative voice outcomes did not significantly differ between patients with a <50% decrease in RLN EMG signal and patients with a 50–90% decrease in RLN EMG signal (Figure 2). Additionally, most NARs with a 50–90% decrease in RLN EMG signal had mild mechanical injury (Table 1). A 50% decrease in RLN EMG signal is reportedly an indicator of imminent RLN injury [24,25]. Schneider et al. [9] reported that a <50% decrease in RLN amplitude is a reliable indicator of intact vocal fold function. To minimize the severity of dysfunction after thyroidectomy, early detection and elimination of the RLN injury mechanism is essential because the cumulative effects of a mild mechanical injury may eventually cause a >90% RLN EMG signal decrease, which indicates a high severity of injury that may be irreversible after surgery [26]. The postoperative voice outcomes substantially differed between a >90% decrease in RLN EMG signal and a <90% decrease in RLN EMG signal (Figure 2). Therefore, in addition to preventing imminent RLN injury (e.g., mild mechanical injury) [27], continuous IONM (C-IONM) can also prevent cumulative increases in the severity of mechanical injury, which can substantially impair postoperative voice outcomes.

The widespread use of energy-based devices (EBDs) in thyroid surgery in recent years has increased the risk of thermal injury in thyroid surgery patients [28]. In the current study, the eight patients with thermal injury included seven (87.5%) patients with an RLN EMG signal decrease >90%. Unlike mild mechanical injury, which usually damages the epineurium and the perineurium, thermal injury tends to cause severe and often irreversible nerve damage in the endoneurium [28]. The thermal injury group had larger impairments in postoperative voice compared to the mechanical injury group (Figure 3). A more detailed analysis revealed that patients with mechanical injury in group C had larger voice impairments compared to patients with mechanical injury in group B (Appendix A). Therefore, during IONM-assisted thyroid surgery, identifying the mechanism of an RLN dysfunction and quantifying its severity are essential. Although patients with RLN dysfunction have a similar incidence of thermal injury and mechanical injury, these two injury types have different effects on thyroidectomy outcome. Surgeons should always consider whether thermal injury has occurred, especially when evaluating new EBD instruments or surgical methods.

Although neuromuscular compensation is well-recognized as a major cause of voice impairment after thyroid surgery [29,30], the current study yielded several insights: (1) if RLN injury is moderate, nerve recovery may occur immediately after surgery, and compensation may result in almost undetectable subjective and objective voice changes; (2) a severe RLN injury may impair the swallowing function. In addition to choking caused by VFM impairment, changes in neuromuscular function associated with swallowing after RLN injury requires further study. (3) Severe RLN injury has a long recovery time. Chiang et al. reported that patients with temporary RLN paralysis did not recover VFM function until three days to as long as four months (mean, 30.7 days) after thyroid surgery [31]. In period-IV, Group C had slight improvements in objective parameters (jitter and shimmer) and in subjective parameters (IVST-T and IVST-V). However, compensation and recovery are much more difficult to achieve in severe RLN injury compared to moderate RLN injury. Therefore, severe RLN injury requires early speech therapy and intervention.

Routine assessment and documentation of subjective and objective voice outcomes after thyroid surgery is recommended in the literature [32]. Since IVST is a subjective voice analysis, preoperative variation by age or gender is expected to be lower than that for objective parameters. Postoperative IVST scores can be a good screening tool for voice impairment and are worthy of further research. For differentiating RLN function, this study revealed that jitter was better than other objective parameters. However, research is needed to develop a novel objective voice analysis parameter that has lower preoperative variation and higher sensitivity to voice change. Additionally, parameters that can be used to predict further voice interventions (i.e., speech therapy or injection laryngoplasty) are also required [33]. Although postoperative subjective voice impairments due to RLN paralysis usually improve over time, they may not return to normal [34]. Since early intervention confers a large benefit in patients with incomplete compensation [35], comprehensive voice assessment is mandatory, and voice therapy and interventions should be performed no longer than six–12 weeks after surgery.

Several limitations of this study should be noted. First, Group B had a higher incidence of bilateral surgery, neck dissection, and malignant pathology results compared to Group A. Nevertheless, the higher incidence of these factors did not result in worse postoperative voice outcomes. Severe RLN dysfunction was still the main factor in postoperative voice impairment. When patients who had less severe RLN dysfunction still suffered from dysphagia and speech disorders, it is necessary to consider factors other than the highest RLN EMG signal decrease. Further research is needed to investigate other contributing factors in postoperative voice impairment. Second, none of the patients in this study received nerve reinnervation or grafting; the voice outcome after these procedures required further research to analyze. Third, the location and other characteristics of thyroid disease (i.e., autoimmune thyroiditis, Graves’ disease), intrathoracic goiters (larger dissection area and more retraction/partial transection of strap muscles), pathologic report and surgical extent, cancer staging, and the injuries to the external branch of superior laryngeal nerve and cricothyroid muscles may have meaningful impacts on voice outcome after thyroidectomy, and these topics should be specifically discussed in future research. Fourth, the duration of follow-up in this study was only three months. Additional studies are needed for a longer observation (e.g., 1-year) of the compensation process. However, the current data were sufficient to reveal important trends in voice parameter changes, and the observed trends support the performing of speech therapy six–12 weeks postoperatively. A future long-term post-thyroidectomy voice study should compare the voice outcomes of patients who received or did receive the voice interventions.

## 5. Conclusions

To the best of our knowledge, this is the first report to discuss the severity and mechanism of RLN dysfunction and postoperative voice in patients who have received monitored thyroidectomy. For optimal voice and swallowing outcomes after thyroid surgery, thermal injury must be avoided, especially when using EBDs, and mechanical injury must be identified early to avoid a more severe dysfunction. Adherence to standard IONM procedures for thyroid surgery is suggested, including standard procedures for acquiring and interpreting intraoperative RLN EMG signals, for identifying and classifying RLN injury mechanisms, for performing laryngeal examinations and comprehensive voice assessments (subjective and objective voice analysis) before and after surgery, and for performing standard follow-up procedures. Future studies may reveal novel voice parameters that can be used as early indicators of the need for voice interventions (e.g., speech therapy or injection laryngoplasty) within six–12 weeks after surgery.

## Figures and Tables

**Figure 1 cancers-13-05379-f001:**
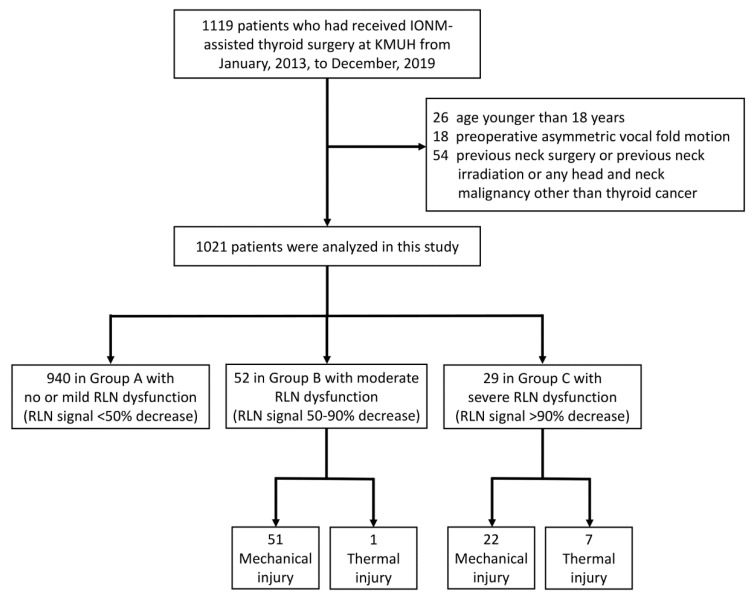
Flowchart of procedure for inclusion and exclusion of patients. Abbreviations: IONM = intraoperative neuromonitoring; RLN = recurrent laryngeal nerve.

**Figure 2 cancers-13-05379-f002:**
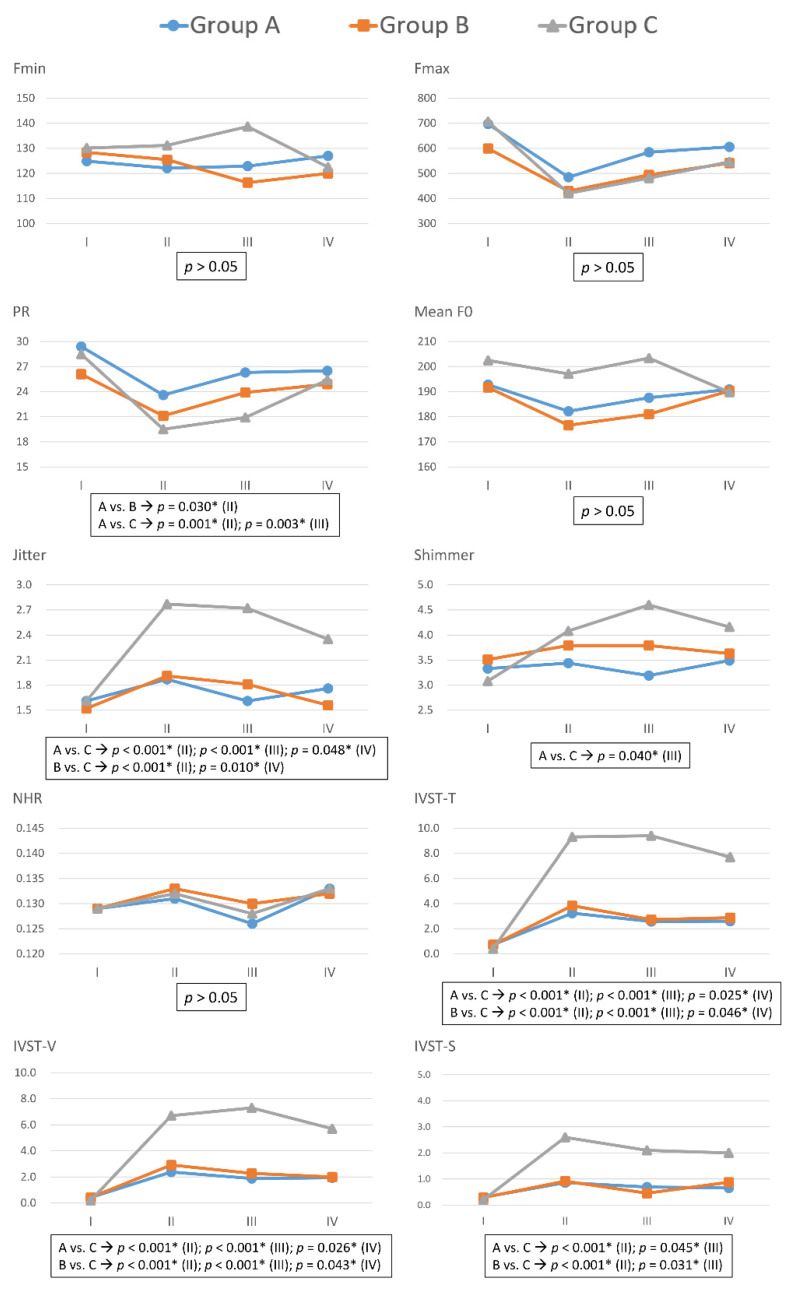
Preoperative and postoperative voice parameters with different severity of recurrent laryngeal nerve dysfunction. In Group A (blue line), B (red line), and C (gray line), post-dissection R2 signals were <50%, 50–90%, and >90% lower than pre-dissection R1 signals, respectively. Fmax = Maximum pitch frequency; Fmin = Minimum pitch frequency; PR = Pitch range; Mean F0 = mean fundamental frequency; NHR = noise-to-harmonic ratio; IVST = Index of Voice and Swallowing Handicap of Thyroidectomy; IVST-T = Total IVST score; IVST-V = IVST score of voice domain score; IVST-S= IVST score of swallowing domain. The units of each voice parameter were as follows: Fmax(Hz), Fmin(Hz), PR(semitone), Jitter(%), Shimmer(%), NHR(value); IVST-T(score); IVST-V(score); IVST-S(score). Period I = Preoperative period (within two months prior to surgery); period II = Immediate postoperative period (median duration of three days; range of one to seven days); period III = Short-term postoperative period (median duration of 12 days; range of seven–30 days); period IV = Long-term postoperative period (median duration of 40 days, range of 30–90 days). Asterisk represents *p* value < 0.05, showed significant difference.

**Figure 3 cancers-13-05379-f003:**
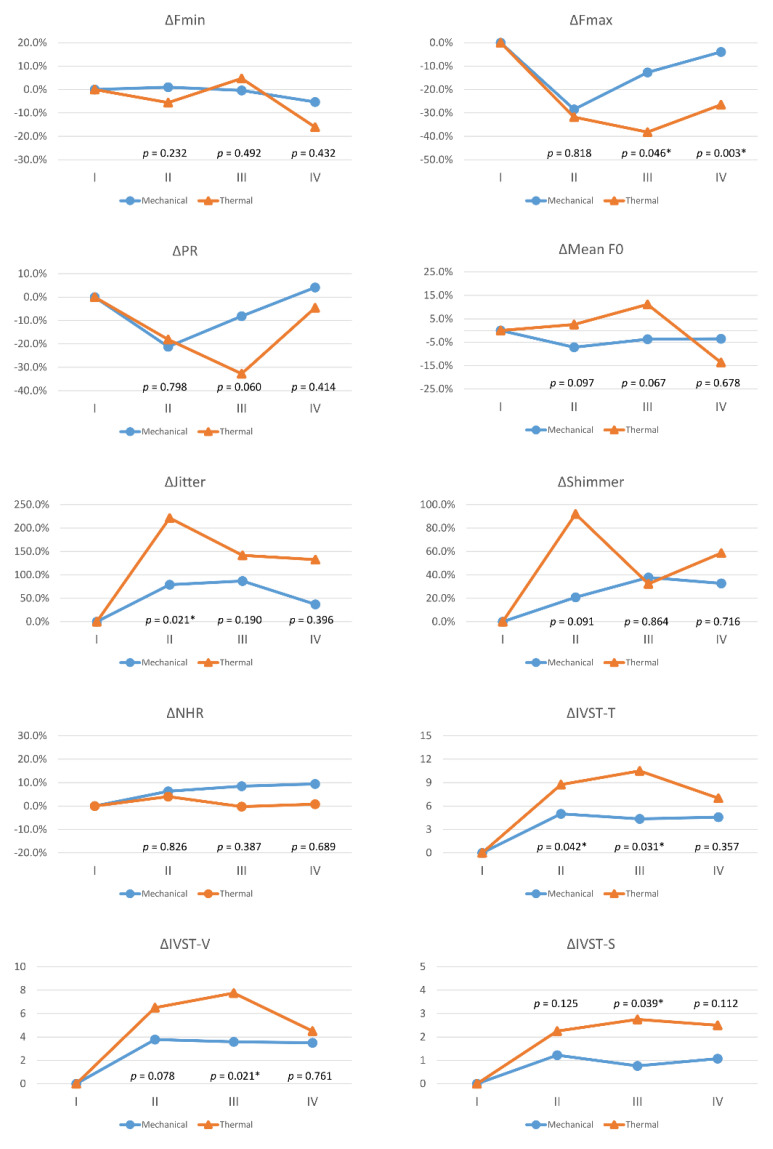
Voice parameter changes (Δ) with different mechanisms of recurrent laryngeal nerve dysfunction. Red lines and blue lines indicate mechanical injury and thermal injury, respectively. Abbreviations for objective/subjective voice parameters and definitions of follow-up periods (I/II/III/IV) are identical to those in Figure 2. The equation for calculating postoperative change in objective voice parameters (Fmax, Fmin, PR, mean F0, jitter, shimmer, NHR) was Δ= (B−A)/A, where the unit is %; the equation for calculating postoperative change in subjective voice parameters (IVST-T, IVST-V, IVST-S) was Δ= B−A, where the unit is score. Values A and B are preoperative and postoperative values, respectively. The preoperative Δ is 0 for all voice parameters. Asterisk represents *p* value < 0.05, showed significant difference.

**Table 1 cancers-13-05379-t001:** Demographic characteristic of patients who received monitored thyroidectomy.

Total 1021 Cases	Group A	Group B	Group C	*p* Value
Number (%)	940 (92.1)	52 (5.1)	29 (2.8)	
Age (mean ± SD)	50.9 ± 13.3	52.2 ± 14.1	52.1 ± 10.4	0.377
Gender				0.968
Male (%)	206 (21.9)	12 (23.1)	6 (20.7)	
Female (%)	734 (78.1)	40 (76.9)	23 (79.3)	
Surgical extent				0.004
Unilateral (%)	254 (27.0)	8 (15.4)	1 (3.4)	
Bilateral (%)	686 (73.0)	44 (84.6)	28 (96.6)	
Pathology				0.017
Benign (%)	610 (64.9)	27 (51.9)	13 (44.8)	
Graves’ disease	37	4	3	
Benign intrathoracic goiter	58	6	3	
Malignant (%)	330 (35.1)	25 (48.1)	16 (55.2)	
Neck dissection				<0.001
Without CND or LND (%)	782 (83.2)	36 (69.3)	13 (44.8)	
With CND, without LND (%)	136 (14.5)	10 (19.2)	13 (44.8)	
With CND and LND (%)	22 (2.3)	6 (11.5)	3 (10.4)	
Injury mechanism and type	1626 NAR	96 NAR	57 NAR	0.006
Mechanical	0 NAR	51 NAR	22 NAR	
Type 1	-	43 NAR	17 NAR	
Type 2	-	8 NAR	5 NAR	
Thermal	0 NAR	1 NAR	7 NAR	
Vocal fold motion				<0.001
Asymmetric during Period-III	0 (0.0)	13 (25.0)	27 (93.1)	
Asymmetric during Period-IV	0 (0.0)	4 (7.7)	18 (62.1)	

Abbreviations: SD = standard deviation; CND = central neck dissection; LND = lateral neck dissection; NAR = nerves at risk. Period-III = Short-term postoperative period (range of 7–30 days); Period-IV = Long-term postoperative period (range of 30–90 days). *p* value <0.05, showed significant difference.

## Data Availability

The original contributions presented in the study are included in the article. Further inquiries can be directed to the corresponding authors.

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
