# Peer review of "How the Severity and Mechanism of Recurrent Laryngeal Nerve Dysfunction during Monitored Thyroidectomy Impact on Postoperative Voice"

_cancers, 2021, doi:10.3390/cancers13215379_

Round 1
Reviewer 1 Report
Thank you for your letter and the changes to the manuscript. I have nothing else to add or comment and look forward to further future studies studying the mechanisms and how to avoid RLN injuries during thyroid surgery.
Reviewer 2 Report
All my commentaries had been replied to in the previous submission. Current submission contains answers to other reviewer's questions, to which I don't feel responsible to reply to. Thus I sustain my previous decision concerning this manuscript
Reviewer 3 Report
Thank you for introducing the changes to the manuscript. All the responses were exhaustive. As a reviewer, I am completely satisfied. In this form the paper is suitable for publication.
Thank you
This manuscript is a resubmission of an earlier submission. The following is a list of the peer review reports and author responses from that submission.
Round 1
Reviewer 1 Report
Huang et al present an impressively large cohort of patients undergoing thyroid surgery that have been assessed regarding their voice pre and post surgery. However, there are several points that I would like to raise;
- The authors conclude that bilateral thyroid surgery is more likely to induce injury to the RLN than unilateral thyroid surgery. Is this not just a result of that in the bilateral surgery, two nerves are at risk in comparison to the unilateral surgery, where only one nerve is at risk. I believe that when studying risk factors for nerve injury, it is better to use nerve a risk instead of a patient.
- The stratification of mechanical and thermal injury is not clear to me, how can you judge the type of injury by a photograph? If this is a new method, then it needs to be described more clearly and include photographs.
- I would like to see stratification also of graves disease, which potential is a risk factor for nerve injury as is also intrathoracic goitres.
- Have the authors controlled for injuries to the EBSLN or direct injuries to the cricothyroid muscles which could also affect the voice?
- In the figures, which are small and hard to read and impossible in black and white printout, I cannot see a measure for the spread of the results for example confidence intervals. Do the dots represent the mean?
Author Response
Author's Response
Dear Reviewer,
We deeply appreciate your comments.
We have revised our manuscript in-line with the comment made.
The followings are our response:
Response to the Reviewer #1
Huang et al present an impressively large cohort of patients undergoing thyroid surgery that have been assessed regarding their voice pre and post surgery. However, there are several points that I would like to raise;
Comment-1:
The authors conclude that bilateral thyroid surgery is more likely to induce injury to the RLN than unilateral thyroid surgery. Is this not just a result of that in the bilateral surgery, two nerves are at risk in comparison to the unilateral surgery, where only one nerve is at risk. I believe that when studying risk factors for nerve injury, it is better to use nerve a risk instead of a patient.
Response:
Thank you for the precious comment. We totally agreed that bilateral surgery increases the nerves at risk (NARs) in comparison to the unilateral surgery. Surgical extent will still be presented in the results, but we removed the description from Abstract and Conclusion section to prevent readers from being misled. (Line 53, 271)
We also thank you point out that, for discussion about injury mechanism, we should use NARs instead of a patient. We modified the description in this article (Line 55, 193-198, 272, 282, 283, Table 1).
Comment-2:
The stratification of mechanical and thermal injury is not clear to me, how can you judge the type of injury by a photograph? If this is a new method, then it needs to be described more clearly and include photographs.
Response:
Thank you for the comment. As you mentioned, mild mechanical injury and lateral thermal spread injury are not visually identifiable and cannot be recorded by photograph. However, in our routine procedure, once the nerve injury was found through IONM, we would stop the surgery to find the location of RLN injury. Magnifying the photo image did allow us to find more injury location than naked eye. We modified our description in Method as, “All exposed RLNs and all visually identifiable injury location (mechanical or thermal) were documented by photograph, and using IONM before and after all the mechanical neural dissections and EBD activations to instantly determine the injury mechanism of RLN.” Thank you again for mentioning this issue.
Comment-3:
I would like to see stratification also of graves disease, which potential is a risk factor for nerve injury as is also intrathoracic goitres.
Have the authors controlled for injuries to the EBSLN or direct injuries to the cricothyroid muscles which could also affect the voice?
Response:
Thank you for your comment. We added the information about Graves’ disease in Table 1. The calculated p value was 0.115 between groups. We completely agree that Graves’ disease and intrathoracic goiters had a meaningful different voice outcome after thyroid surgery compared to benign disease. Although the injury of EBSLN and CTM has less impact on voice outcome than the RLN injury, we believe that investigation of EBSLN/CTM is as important as RLN for understanding the voice issues after thyroidectomy. To clarify these two issues, the completely different research designs are required, and we are also willing to demonstrate our results in future researches. Thank you again for your suggestion and mentioning the important issues, we added the description in Discussion/Limitation section as “Third, the location and other characteristics of thyroid disease (i.e. autoimmune thyroiditis, Graves’ disease, intrathoracic goiters), pathologic report and surgical extent, cancer staging, and the injuries to the external branch of superior laryngeal nerve and cricothyroid muscles may have the meaningful impacts on voice outcome after thyroidectomy, these topics should be specifically discussed in future research.”
Comment-4:
In the figures, which are small and hard to read and impossible in black and white printout, I cannot see a measure for the spread of the results for example confidence intervals. Do the dots represent the mean?
Response:
Thank you for your precious comment. We modified the Figure 2 and Figure 3 to enlarge the font and to modify the mark. We only show the mean value in the figure, and the standard deviation was shown in supplemental table to avoid too much confusion on the figure. Hope you can understand our intentions, if a confidence interval is necessary, we are also willing to add it in.
We thank you for your valued comments and suggestions, which we feel substantially improve our manuscript, and hope that the revisions meet with your approval.
Sincerely,
Tzu-Yen Huang, M.D.
Department of Otorhinolaryngology–Head and Neck Surgery, Kaohsiung Medical University Hospital, Kaohsiung Medical University, Kaohsiung, Taiwan.
Address: 100TzYou 1st Road, Kaohsiung 807, Taiwan.
E-mails: tyhuang.ent@gmail.com
Sheng-Hsuan Lin, M.D., Sc.D., Sc.M
Institute of Statistics, National Yang Ming Chiao Tung University, Hsinchu, Taiwan
Address: 1001 University Road, Hsinchu 300, Taiwan.
E-mails: shenglin@nctu.edu.tw
(On behalf of all coauthors)
Sep 21, 2021

Reviewer 2 Report
The study by Tzu-Yen Huang et al. deals with a very serious problem of injury to the RL nerve. The study is well written and interesting to read. I like the fact that the study is based on a large group of patients. I have a few suggestions on how to improve the work:
Line 40 - IONM – please explain abbreviation
Line 55 - period-IV – please explain
Line 118 - four-step (V1-R1-R2-V2) procedure – please provide a short explanation
Line 136, 137 – CND, LND – please explain abbreviations
Line 185, 186 - type 1 injury, type 2 injury – please explain
Line 205 – PR – please explain
Line 339 - A future long-term post-thyroidectomy voice study should include discussion of voice interventions and their effects – perhaps it would be possible to describe interventions that helped to regain or improve voice in the group of patients described by the authors?
Author Response
Author's Response
Dear Reviewer,
We deeply appreciate your comments.
We have revised our manuscript in-line with the comment made.
The followings are our response:
Response to the Reviewer #2
The study by Tzu-Yen Huang et al. deals with a very serious problem of injury to the RL nerve. The study is well written and interesting to read. I like the fact that the study is based on a large group of patients. I have a few suggestions on how to improve the work:
Comment-1:
Line 40 - IONM – please explain abbreviation
Line 55 - period-IV – please explain
Line 118 - four-step (V1-R1-R2-V2) procedure – please provide a short explanation
Line 136, 137 – CND, LND – please explain abbreviations
Line 185, 186 - type 1 injury, type 2 injury – please explain
Line 205 – PR – please explain
Line 339 - A future long-term post-thyroidectomy voice study should include discussion of voice interventions and their effects – perhaps it would be possible to describe interventions that helped to regain or improve voice in the group of patients described by the authors?
Response:
Thank you for your comments, we have modified the descriptions as your suggestion. About Line 339, we modified our description as “A future long-term post-thyroidectomy voice study should compare the voice outcomes of patients who received or not received the voice interventions.” Thank you for your suggestion to make our narrative more precise.
We thank you for your valued comments and suggestions, which we feel substantially improve our manuscript, and hope that the revisions meet with your approval.
Sincerely,
Tzu-Yen Huang, M.D.
Department of Otorhinolaryngology–Head and Neck Surgery, Kaohsiung Medical University Hospital, Kaohsiung Medical University, Kaohsiung, Taiwan.
Address: 100TzYou 1st Road, Kaohsiung 807, Taiwan.
E-mails: tyhuang.ent@gmail.com
Sheng-Hsuan Lin, M.D., Sc.D., Sc.M
Institute of Statistics, National Yang Ming Chiao Tung University, Hsinchu, Taiwan
Address: 1001 University Road, Hsinchu 300, Taiwan.
E-mails: shenglin@nctu.edu.tw
(On behalf of all coauthors)
Sep 21, 2021

Reviewer 3 Report
A very interesting, well-illustrated retrospective paper analyzing a group of over 1000 patients undergoing primary uni- or bilateral thyroidectomy using intraoperative neuromonitoring in terms of objective and subjective postoperative voice assessment and the cause of recurrent laryngeal nerve injury.
The authors reported that this was the first such a comprehensive analysis.
Nevertheless, I have some questions and concerns, which are as follows:
In the Methods section, the Authors state that "all exposed RLNs and their injury mechanisms (mechanical or thermal) were documented by photograph”. Please explain in more detail this statement. What does ”by photograph” mean?
The Authors stated that studies on vocal cord movement were "performed monthly until recovery". Does this mean that all paresis were transient/temporary? This is not clear from the data presented as vocal cord movement and symmetry abnormalities were observed during periods III and IV. Please clarify.
According to the Authors, “the patients were classified into three groups according to the RLN side with the most severe dysfunction: Group A included patients had no/mild RLN dysfunction with <50% RLN signal decrease; Group B included patients had moderate RLN dysfunction 124 with 50-90% RLN signal decrease; Group C included patients had severe RLN dysfunction with >90% RLN signal decrease” and that none of the patients with bilateral thyroidectomy presented with an RLN signal decrease >50% on both sides.
However, patients may have had less severe signal decrease, which may have influenced other disorders such as dysphagia or speed disorders.
How were such patients classified – based on the highest decrease?
I wonder if only patient assessment is sufficient. Shouldn't the number of nerves exposed be evaluated? In bilateral surgical procedures, two nerves are exposed. By definition, more extensive surgical procedures result in the possibility of more speech disorders or dysphagia.
More extensive surgeries involve patients with malignant tumors.
It is the diagnosis of malignant neoplasm and the extent of surgery (often CND and/or LND) that results in a greater risk of nerve injury, not the other way around.
An highly interesting and definitely valuable observation of this study is the significance of thermal injury. In the era of modern electrocoagulation equipment or harmonic knives, surgeons should be made aware of the fact that these tools should still be avoided in the close proximity of the nerve, as thermal injury is much more severe for patients.
As clearly stated by the Authors “therefore, for optimal voice and swallowing outcomes after thyroidectomy, avoiding thermal injury is mandatory, and mechanical injury must be identified early to avoid a more severe dysfunction”.
The authors rightly listed the limitations of the study.
The study assessed only the influence of RLN and not other factors e.g. the significance of injury to the external branch of superior laryngeal nerve.
I agree that a longer follow-up is also necessary.
For me, an additional disadvantage of the study is the heterogeneity of patient groups. Patients with benign neoplasms should be evaluated separately from those with malignant ones in terms of the stage of the disease, e.g. TNM classification.
Do the authors have any data on the percentage of patients with autoimmune thyroiditis in particular groups? Under inflammatory conditions, nerve dissection is often more difficult. It requires dissection from lymph nodes and inflammatory tissues.Are the Authors able to complete this data?
The conclusions are valid. The study is valuable, interesting, innovative, comprehensive, using subjective and objective tools for postoperative evaluation of voice disorders on a large group of patients and hence deserves to be published.
Thank you
Author Response
Author's Response
Dear Reviewer,
We deeply appreciate your comments.
We have revised our manuscript in-line with the comment made.
The followings are our response:
Response to the Reviewer #3
A very interesting, well-illustrated retrospective paper analyzing a group of over 1000 patients undergoing primary uni- or bilateral thyroidectomy using intraoperative neuromonitoring in terms of objective and subjective postoperative voice assessment and the cause of recurrent laryngeal nerve injury.
The authors reported that this was the first such a comprehensive analysis.
Nevertheless, I have some questions and concerns, which are as follows:
Comment-1:
In the Methods section, the Authors state that "all exposed RLNs and their injury mechanisms (mechanical or thermal) were documented by photograph”. Please explain in more detail this statement. What does ”by photograph” mean?
Response:
Thank you for your comment. Mild mechanical injury and lateral thermal spread injury are not visually identifiable and cannot be recorded by photograph. However, in our routine procedure, once the nerve injury was found through IONM, we would stop the surgery to find the location of RLN injury. Magnifying the photo image did allow us to find more injury location than naked eye. We modified our description in Method as, “All exposed RLNs and all visually identifiable injury location (mechanical or thermal) were documented by photograph, and using IONM before and after all the mechanical neural dissections and EBD activations to instantly determine the injury mechanism of RLN.” Thank you again for mentioning this issue.
Comment-2:
The Authors stated that studies on vocal cord movement were "performed monthly until recovery". Does this mean that all paresis were transient/temporary? This is not clear from the data presented as vocal cord movement and symmetry abnormalities were observed during periods III and IV. Please clarify.
Response:
Thank you for your comment, we totally agree with you. Because the data was collected until 3 months after surgery, the abnormal VFM in period IV is still not completely equivalent to permanent vocal fold paralysis (usually defined as 6 months or 12 months), so we described the VFM status during periods III and IV instead of using transient/permanent VFM. For avoiding confusion, we added the description in Methods section as, “Patients who had asymmetric VFM 3 months after surgery would be referred to voice interventions, and the follow-up time will depend on the intervention patient received.” Thank you again for your suggestion, which really helps us to clarify this issue.
Comment-3:
According to the Authors, “the patients were classified into three groups according to the RLN side with the most severe dysfunction: Group A included patients had no/mild RLN dysfunction with <50% RLN signal decrease; Group B included patients had moderate RLN dysfunction with 50-90% RLN signal decrease; Group C included patients had severe RLN dysfunction with >90% RLN signal decrease” and that none of the patients with bilateral thyroidectomy presented with an RLN signal decrease >50% on both sides.
- However, patients may have had less severe signal decrease, which may have influenced other disorders such as dysphagia or speed disorders. How were such patients classified – based on the highest decrease?
- I wonder if only patient assessment is sufficient. Shouldn't the number of nerves exposed be evaluated? In bilateral surgical procedures, two nerves are exposed.
- By definition, more extensive surgical procedures result in the possibility of more speech disorders or dysphagia. More extensive surgeries involve patients with malignant tumors. It is the diagnosis of malignant neoplasm and the extent of surgery (often CND and/or LND) that results in a greater risk of nerve injury, not the other way around.
- The study assessed only the influence of RLN and not other factors e.g. the significance of injury to the external branch of superior laryngeal nerve.
- For me, an additional disadvantage of the study is the heterogeneity of patient groups. Patients with benign neoplasms should be evaluated separately from those with malignant ones in terms of the stage of the disease, e.g. TNM classification.
- Do the authors have any data on the percentage of patients with autoimmune thyroiditis in particular groups? Under inflammatory conditions, nerve dissection is often more difficult. It requires dissection from lymph nodes and inflammatory tissues.Are the Authors able to complete this data?
Response:
Thank you for your positive response and provide us with many valuable suggestions.
For comment 4-(1), we totally agree with you and added description as “When patients who had less severe RLN dysfunction still suffered from dysphagia and speech disorders, it is necessary to consider factors other than the highest RLN EMG signal decrease. Further research is needed to investigate other contributing factors in postoperative voice impairment.” In addition, we have added the description about “other contributing factors” below based on your suggestion.
For comment 4-(2), we totally agreed that bilateral surgery increases the nerves at risk (NARs) in comparison to the unilateral surgery. Surgical extent will still be presented in the results, but we removed the description from Abstract and Conclusion section to prevent readers from being misled. (Line 53, 271) We also thank you point out that, for discussion about injury mechanism, we should use NARs instead of a patient. We modified the description in this article (Line 55, 193-198, 272, 282, 283, Table 1).
For comment 4-(3), the issue about pathology reports (benign vs. malignancy) and extent of surgery (CND and/or LND). For comment 4-(4), the issue about EBSLN/CTM. Although the injury of EBSLN and CTM has less impact on voice outcome than the RLN injury, we believe that investigation of EBSLN/CTM is as important as RLN for understanding the voice issues after thyroidectomy. For comment 4-(5), the issue about cancer staging. To evaluate the voice impact of above factors, the completely different research designs are required, and we are also willing to demonstrate our results in future researches.
For comment 4-(6), the issue about Graves’ disease. We added the information about Graves’ disease in Table 1. The calculated p value was 0.115 between groups. We completely agree that autoimmune thyroiditis and Graves’ disease had a meaningful different voice outcome after thyroid surgery compared to benign disease.
To clarify the issues from the comment 4, we added the description in Discussion/Limitation section as “Third, the location and other characteristics of thyroid disease (i.e. Graves’ disease, in-trathoracic goiters), pathologic report and surgical extent, cancer staging, and the injuries to the external branch of superior laryngeal nerve and cricothyroid muscles may have the meaningful impacts on voice outcome after thyroidectomy, these topics should be specifically dis-cussed in future research.” Thank you again for your suggestion and mentioning the important issues
We thank you for your valued comments and suggestions, which we feel substantially improve our manuscript, and hope that the revisions meet with your approval.
Sincerely,
Tzu-Yen Huang, M.D.
Department of Otorhinolaryngology–Head and Neck Surgery, Kaohsiung Medical University Hospital, Kaohsiung Medical University, Kaohsiung, Taiwan.
Address: 100TzYou 1st Road, Kaohsiung 807, Taiwan.
E-mails: tyhuang.ent@gmail.com
Sheng-Hsuan Lin, M.D., Sc.D., Sc.M
Institute of Statistics, National Yang Ming Chiao Tung University, Hsinchu, Taiwan
Address: 1001 University Road, Hsinchu 300, Taiwan.
E-mails: shenglin@nctu.edu.tw
(On behalf of all coauthors)
Sep 21, 2021

Round 2
Reviewer 1 Report
Thank you for your letter and the revised manuscript, that has improved since the research design now describes nerves at risk rather than number of patients. I would, however, have wish for a more thorough revision of the manuscript according to my former comments. As previously commented, the number of patients included in the study and the voice evaluation follow up are impressive and deserve a better controlled, designed study and improved presentation of the results section. I look forward to see your future results.